# Defining murine monocyte differentiation into colonic and ileal macrophages

Mor Gross-Vered[1], Sébastien Trzebanski[1], Anat Shemer[1], Biana Bernshtein[1], Caterina Curato[1], Gil Stelzer[2], Tomer-Meir Salame[2], Eyal David[1], Sigalit Boura-Halfon[1], Louise Chappell-Maor[1], Dena Leshkowitz[2], Steffen Jung[1]*

[1]Department of Immunology, Weizmann Institute of Science, Rehovot, Israel; [2]Bioinformatics Unit, Life Science Core Facilities, Weizmann Institute of Science, Rehovot, Israel

**Abstract** Monocytes are circulating short-lived macrophage precursors that are recruited on demand from the blood to sites of inflammation and challenge. In steady state, classical monocytes give rise to vasculature-resident cells that patrol the luminal side of the endothelium. In addition, classical monocytes feed macrophage compartments of selected organs, including barrier tissues, such as the skin and intestine, as well as the heart. Monocyte differentiation under conditions of inflammation has been studied in considerable detail. In contrast, monocyte differentiation under non-inflammatory conditions remains less well understood. Here we took advantage of a combination of cell ablation and precursor engraftment to investigate the generation of gut macrophages from monocytes. Collectively, we identify factors associated with the gradual adaptation of monocytes to tissue residency. Moreover, comparison of monocyte differentiation into the colon and ileum-resident macrophages revealed the graduated acquisition of gut segment-specific gene expression signatures.

## Introduction

The recent past has seen major advance in our understanding of the diverse origins of tissue macrophages, as well as their discrete maintenance strategies. Macrophages were shown to arise from three distinct developmental pathways that differentially contribute to the tissue compartments in the embryo and adult (*Ginhoux and Guilliams, 2016*). In the mouse, embryonic tissue macrophages first develop from primitive macrophage progenitors that originate in the yolk sac (YS). In the brain, YS-macrophage-derived macrophages persist throughout adulthood, while in most other tissues these cells are replaced by fetal monocytes that derive from multi-potent erythro-myeloid progenitors (EMP). Definitive hematopoiesis commences from E10.5 with the generation of hematopoietic stem cells (HSC) that first also locate to the fetal liver, but eventually seed the bone marrow (BM) to maintain adult hematopoiesis. Most EMP-derived tissue macrophage compartments persevere throughout adulthood without major input from HSC-derived cells; however, in certain barrier tissues, as well as selected other organs, like the heart, embryonic macrophages are progressively replaced by HSC-derived cells involving a blood monocyte intermediate (*Varol et al., 2015*).

Monocytes are continuously generated in the BM involving a sequence of developmental intermediates, before extravasation into the circulation (*Ginhoux and Jung, 2014*; *Mildner et al., 2016*). Once in the blood, murine classical Ly6C$^+$ monocytes have a limited half-life (*Yona et al., 2013*). On demand, Ly6C$^+$ monocytes can be rapidly recruited to sites of injury and challenge, where they complement tissue-resident macrophages and dendritic cells. In absence of challenge, some Ly6C$^+$ monocytes give rise to vasculature-resident Ly6C$^-$ cells, which patrol the vessel walls (*Auffray et al., 2007*). These cells display macrophage characteristics including extended life spans (*Yona et al., 2013*; *Mildner et al., 2017*). In addition, Ly6C$^+$ monocytes replenish the above-mentioned selected

*For correspondence:
s.jung@weizmann.ac.il

Competing interests: The authors declare that no competing interests exist.

**eLife digest** Macrophage cells play a crucial role in keeping the body free of disease-causing microbes and debris. They surveille the tissues, detect and clear infections, and tidy up dead cells. Most internal organs contain a population of macrophages that move into the organ during development and then persist throughout an organism's life. However, tissues in contact with the outside world, such as the gut, need a constant supply of fresh macrophages. This supply depends on immune cells called monocytes moving into these tissues from the blood and maturing into macrophages when they arrive.

The macrophages in the gut have a challenging job to do. They need to be able to detect infections amongst healthy gut bacteria and foreign food particles. Macrophages from other tissues would overreact if they encountered this complicated environment, but gut macrophages learn to tolerate their surroundings by switching genes on and off as they mature. The exact combination of genes macrophages in the gut use depends on whether they are in the small or large intestine, which have different anatomies and resident microbes.

To understand how monocytes mature into macrophages in the gut, previous studies have focused on what happens during an infection. However, it remains unclear how monocytes develop into mature gut macrophages in the healthy gut. To address this question, Gross-Vered et al. have looked at mice in which gut macrophages can be killed when a drug is applied. This made it possible to replace the mice's own macrophages with fluorescently labelled cells derived from monocytes.

Fluorescent monocytes were introduced into the bloodstream and arrived in the small and large intestine after the drug had been administered. Gross-Vered et al. then collected cells derived from these labelled monocytes and examined the genes that they were using. This revealed that once the monocytes entered the gut they began sensing their new environment and switching thousands of genes on and off. These changes happened rapidly at first and continued more gradually as the macrophages matured. Comparing the fluorescent macrophages from the small and large intestines revealed many similarities, but there were also hundreds of genes that differed. In the small intestine, macrophages switched on genes involved in catching and consuming bacteria, whereas macrophages in the large intestine, which has more resident healthy bacteria, turned on fewer of these bacteria-eating genes.

Inflammatory bowel disorders like ulcerative colitis and Crohn's disease both involve gut macrophages. Comparing the genes that macrophages use in the healthy and diseased gut may reveal information about these disorders. For example, ulcerative colitis only affects the large intestine, so understanding how and why the monocytes mature differently there could shed light on new ways to treat the disease.

---

steady state tissue macrophage compartments, including the gut and skin (*Ginhoux and Jung, 2014*). Given their mobility, plasticity and key role in pathologies, the manipulation of monocytes and their differentiation could bear considerable therapeutic potential. However, monocyte differentiation into tissue resident cells remains incompletely understood.

Gut macrophages, which reside in the connective tissue underlying the gut epithelium, the *lamina propria*, are considered key players for the maintenance of intestinal homeostasis. As such, they constantly sense their environment and respond to the unique microbiota and food challenge (*Bain and Mowat, 2011*; *Zigmond and Jung, 2013*). Recent studies revealed that monocyte-derived *lamina propria* macrophages comprise in mice two populations, that is short-lived cells and long-lived cells with self-renewing capacity (*Shaw et al., 2018*); the latter population might also include remnants of embryonic populations, as could additional intestinal long-lived macrophage populations that reside near blood vessels, nerves and in the Peyer's Patches (*De Schepper et al., 2018*). Evidence for macrophages with different half-lives is also emerging for the human small intestine (*Bujko et al., 2018*). Collectively, these findings highlight the existence of considerable macrophage heterogeneity, not only between different organs, but also within given tissues.

Monocyte differentiation into intestinal macrophages involves phenotypic changes with respect to Ly6C, CD64 and MHCII expression, a sequence described as 'monocyte waterfall'

(*Tamoutounour et al., 2012*). Mature steady-state gut macrophages tolerate the commensal microbiota and food antigens (*Bain and Mowat, 2011*; *Zigmond and Jung, 2013*). Their relative unresponsiveness is thought to rely on regulatory circuits that balance the expression of pro- and anti-inflammatory gene products, such as cytokines and molecules participating in pattern recognition receptor signaling cascades.

Macrophages located in different tissues display characteristic enhancer landscapes and gene expression profiles, which have been attributed to the exposure of instructing factors of the microenvironment they reside in *Amit et al. (2016)*. Gut segments display distinct anatomy, function and microbiota load (*Mowat and Agace, 2014*) and macrophages of small and large intestine hence also likely differ. Despite the known distinct susceptibility of the colon and ileum to pathology, such as to the IL10R deficiency (*Glocker et al., 2009*; *Zigmond et al., 2014*) and in ulcerative colitis, so far no comparative analysis of their macrophages has been reported. Likewise, our general understanding of monocyte-derived tissue resident macrophages remains scarce and is largely restricted to settings of inflammation.

Here we investigated monocyte differentiation into intestinal macrophages in the small and large intestine. Using adoptive monocyte transfers into macrophage-depleted recipients (*Varol et al., 2007*; *Varol et al., 2009*), we synchronized the macrophages in terms of development, isolated colonic and ileal macrophages and subjected them to transcriptome profiling. Our data establish the distinct identities of gut segment resident macrophages and shed light on the kinetics and gradual gene expression of specific factors for their establishment of their identities.

## Results

### Monocyte transcriptomes acquired during differentiation into ileal and colonic gut macrophages

Tissue macrophages display distinct gene expression profiles and enhancer landscapes (*Amit et al., 2016*). This holds also for intestinal macrophages residing in small and large intestine. Transcriptomes of $Ly6C^+$ BM monocytes, that is the macrophage precursors, and transcriptomes of their progeny in colon and ileum displayed 6200 genes differentially expressed at least 2-fold across all analyzed data sets out of a total of 12345 detected genes (*Figure 1—figure supplement 1A–C*). 2255 genes were expressed in monocytes and down-regulated in macrophages (cluster I). Conversely, cluster II comprised genes whose expression was absent from monocytes, but shared by both small and large intestinal macrophages. Finally, 1087 and 987 genes were either preferentially or exclusively expressed in ileal or colonic macrophages, respectively.

To further characterize adult monocyte-derived gut macrophages, we took advantage of an experimental system involving monocyte engraftment of macrophage-depleted animals (*Varol et al., 2007*; *Varol et al., 2009*). Analysis of transferred cells at different intervals from engraftment allows the study of intestinal macrophage development over time, since monocyte differentiation is synchronized. For the cell ablation we used [CD11c-DTR > WT] BM chimeras, in which diphtheria toxin (DTx) receptor (DTR) transgenic intestinal macrophages can be conditionally ablated by DTx injection (*Varol et al., 2007*; *Varol et al., 2009*) (*Figure 1A*). Two days prior to monocyte transfer, DTx was applied to the recipients to clear their intestinal macrophage niche and mice were subsequently treated with DTx every second day. The monocyte graft was isolated from BM of $Cx_3cr1^{GFP/+}$ mice (*Jung et al., 2000*) and defined as $CD117^-$ $CD11b^+$ $CD115^+$ $Ly6C^+$ $GFP^{int}$ cells; donor animals also harbored an allotypic marker (CD45.1) (*Figure 1A*, *Figure 1—figure supplement 1A*). Grafted cells could be visualized in recipient gut tissue and underwent expansion, as reported earlier (*Varol et al., 2009*) (*Figure 1—figure supplement 2*). Intestinal tissues of recipient mice were harvested on day 4, 8 and 12 post engraftment (*Figure 1A*) to isolate graft-derived macrophages according to CD45.1 and $CX_3CR1$/GFP expression; these cells were $CD11b^+$ $CD64^+$ $Ly6C^-$ and mainly $CD11c^{hi}$ (*Figure 1B*). Of note, specifics of the system preclude harvest of graft-derived cells at later time points (*Varol et al., 2009*). The monocyte graft and colonic and ileal macrophage populations were subjected to bulk RNA-seq using MARS-Seq technology (*Jaitin et al., 2014*).

Gene expression profiling revealed robust changes already at day four post engraftment; changes appeared to be tissue- rather than time-specific, with the exception of two day 4 samples of ileal macrophages, which clustered with the colonic samples (*Figure 1C*, *Figure 1—figure supplement*

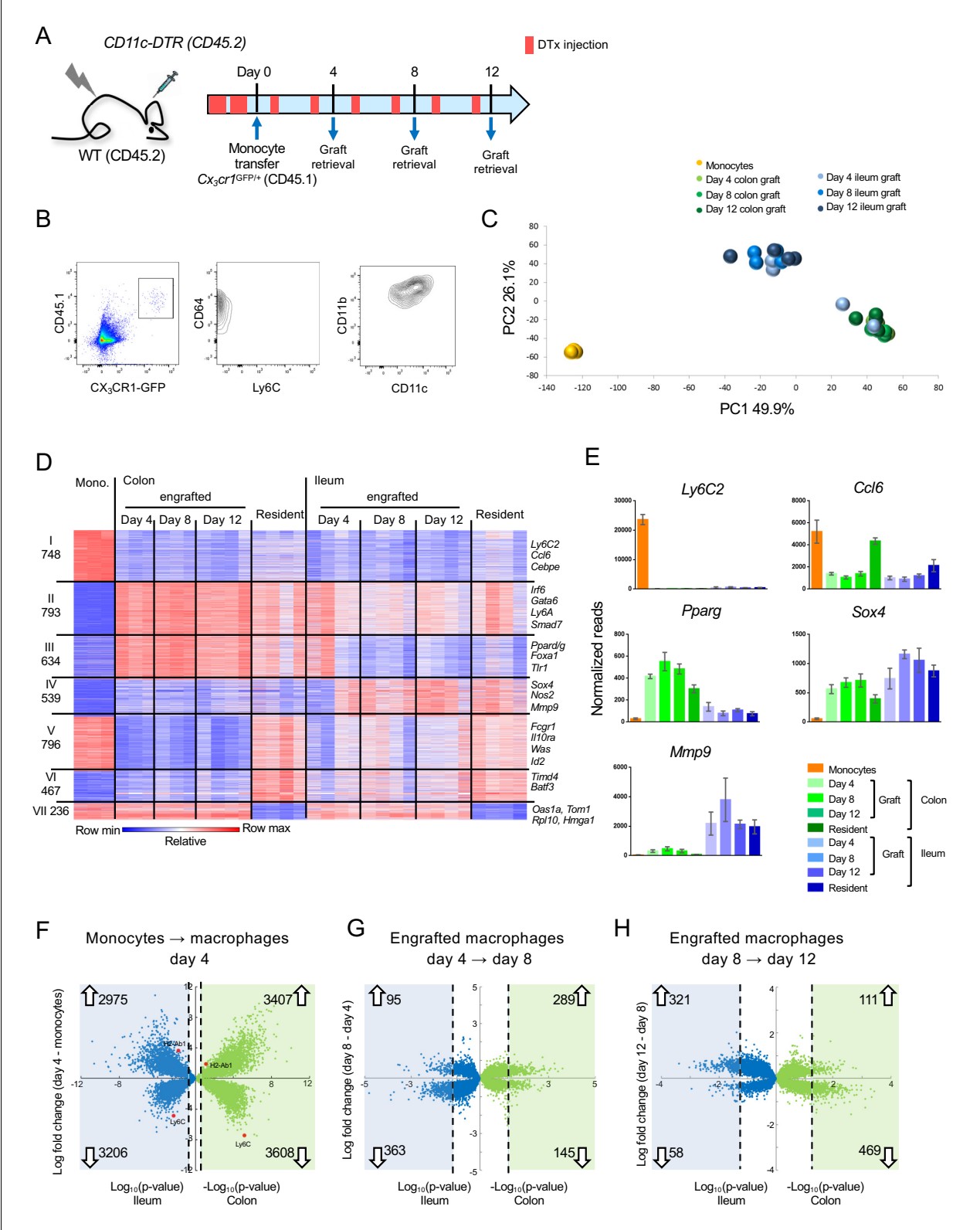

**Figure 1.** Transcriptome analysis of CX$_3$CR1$^+$ monocyte graft - derived colonic and ileal macrophages isolated from macrophage depleted animals. (**A**) Experimental protocol. Briefly, [CD11c-DTR > C57BL/6] BM chimeras were treated as indicated with DTx. 2 × 10$^6$ CD117$^-$ CD11b$^+$ CD115$^+$ Ly6C$^+$ GFP$^{int}$ BM monocytes isolated from *Cx$_3$cr1$^{GFP/+}$* mice were injected intravenously. Macrophages were sorted from the colon and ileum at days 4, 8 and 12 post-transfer. Experiment was repeated three times, total 3–4 samples from each tissue at each time point. (**B**) Graft-derived macrophages were sorted

*Figure 1 continued on next page*

*Figure 1 continued*

at days 4, 8 and 12 post transfer based on CD45.1 and CX³CR1 (GFP) expression. GFP⁺ cells also express CD64, CD11b and variable levels of CD11c, and lacked Ly6C expression. (C) Principal component analysis (PCA) of transcriptomes of BM monocytes and grafted cells from colon and ileum at 4, 8 and 12 days post-transfer. Analysis performed in MATLAB. (D) Expression heat map of 4213 genes that show at least four fold change across all samples in the dataset, divided to seven clusters by the unbiased K-means algorithm in MATLAB. (*Supplementary file 1*, data sets 1,2, 5–11). (E) Representative genes from heat map in (D). (F–H) Double volcano plots of genes that change from monocytes to day 4, day 4 to day 8 and day 8 to day 12 in the colon (green dots) and ileum (blue dots). Blue and green squares indicate genes that are significantly (Student's T-test, p-value<0.05) changed between the different time points. Numbers within blue and green squares represent the number of genes in the square, namely genes that significantly change. Arrows indicate up –or down-regulation.

The online version of this article includes the following figure supplement(s) for figure 1:

**Figure supplement 1.** Comparison of gene expression in monocytes and resident colonic and ileal macrophages.
**Figure supplement 2.** Analysis of host tissue for graft-derived cells.
**Figure supplement 3.** Pathway analysis of graft-derived cells.
**Figure supplement 4.** Analysis of expression of pro-inflammatory genes in adoptive transfer model and DSS-induced colitis model.

*3A*). Transcriptomes of the retrieved engrafted macrophages lacked expression of pro-inflammatory genes, such as *Saa3, Lcn2, Il1b, Il6* and *Tnf*, as opposed to cells retrieved from colitic animals treated with Dextran Sulfate Sodium (DSS) (*Okayasu et al., 1990*) (*Figure 1—figure supplement 4*). This supports the earlier notion that the cell ablation results in a transient tissue response, but the monocyte transfer system mimics cell differentiation close to steady-state conditions (*Varol et al., 2009*).

Comparative transcriptome analysis of grafted monocytes, their progeny retrieved from the recipient intestines, and resident ileal and colonic macrophages that were independently retrieved from $Cx_3cr1^{GFP/+}$ mice revealed 4213 differentially expressed genes (DEG) (>4 fold differences in any pairwise comparison among a total of 12878 genes) (*Figure 1D*). 748 genes were exclusively expressed in the monocyte graft, including *Ly6c2, Ccl6* and *Cebpe* (Cluster I) (*Figure 1D,E*). 793 genes shared expression in graft-derived cells and gut macrophages in colon and ileum (Cluster II). These included *Irf6, Gata6, Ly6A* and *Smad7*. Cluster III comprised 634 genes that were either exclusively or preferentially expressed in colonic macrophages, such as *Pparg, Foxa1* and *Tlr1* (*Figure 1E*). Cluster IV comprised 539 genes exclusively or preferentially expressed in ileal macrophages, such as *Nos2, Sox4* and *Mmp9* (*Figure 1E*). Metascape analysis (*Zhou et al., 2019*) highlighted that genes associated with cluster II, II and IV were associated with distinct pathways, particularly with respect to epithelial cell communication and cell metabolism (*Figure 1—figure supplement 3B*). We also identified genes that differed in expression between engrafted and resident macrophages. Specifically, 796 and 467 genes displayed high or low expression levels in monocytes, respectively, were highly expressed in resident colonic macrophages, but either low or absent in the graft-derived cells (Cluster V and VI). Finally, cluster VII comprised 236 genes that were expressed in monocytes, and down-regulated in resident gut macrophages, but not in the grafted cells.

Volcano plot analysis revealed that early monocyte differentiation (graft vs day 4) was characterized by abundant changes in gene expression in both colon and ileum. In the colon 3407 genes were up- and 3608 genes were down-regulated; in the ileum 2975 genes were up- and 3206 genes were down-regulated, with up to 10 fold change (*Figure 1F*). Later time points (day 4 to day 8 and day 8 to day 12) were characterized by less pronounced alterations, both with respect to the number of DEG and their fold change (*Figure 1G,H*). In the colon 289 genes (66% of all significantly-changed genes) were induced between day 4 and 8 and only 111 genes (19%) were up-regulated between day 8 and 12. In the ileum this trend was reversed, with 95 genes (21%) up-regulated between day 4 and 8, but 321 genes (85%) induced between day 8 and 12. Collectively, this establishes that monocytes that enter distinct gut segments rapidly acquire characteristic transcriptomic signatures.

## Transcriptomes of engrafted cells differ from resident macrophages

Our experimental set up allows us to focus on cells that entered the gut in a defined time window. Interestingly, even by day 12, monocyte graft-derived cells differed in expression profiles when compared to resident ileal and colonic macrophages (*Figure 2A*). Differences included genes absent from, and exclusively expressed by engrafted cells (*Figure 1D* clusters V - VII, *Figure 2B–D*). These differences might be attributable to incomplete macrophage maturation or to the recently reported heterogeneity of the intestinal macrophage compartment (*Shaw et al., 2018*; *De Schepper et al.,*

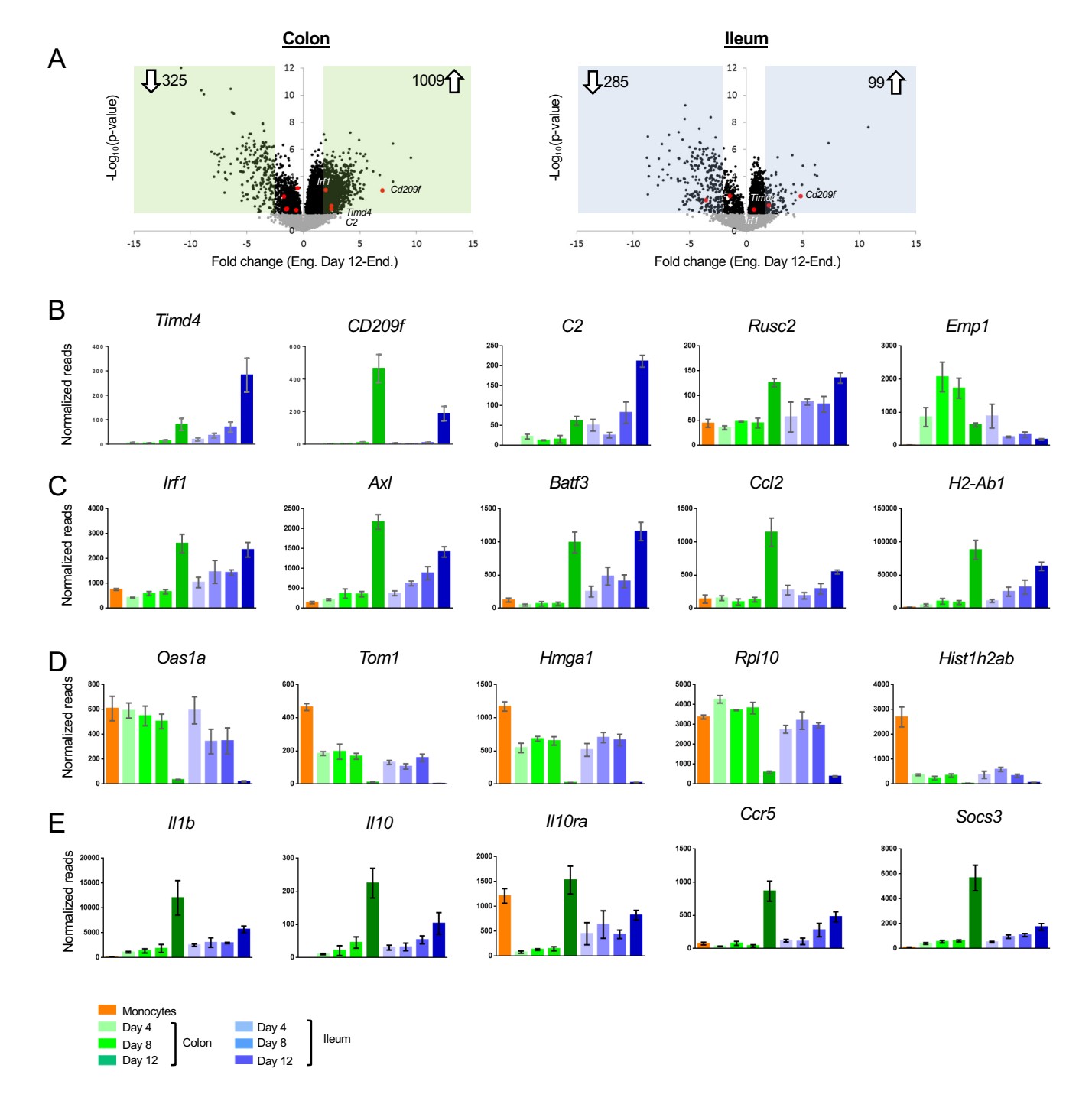

**Figure 2.** Differences between acute monocyte graft -derived macrophages and resident intestinal macrophages. (**A**) Volcano plots of genes that change from engrafted macrophages at day 12 and resident macrophages in the colon (left) and ileum (right). Gray dots represent genes that do not significantly change (p-value>0.05), black dots represent genes that significantly change (p-value<0.05). Green and blue squares mark genes with at least 2-fold log fold change (equals 4-fold read change). Numbers inside squares indicate number of genes in square, namely genes that significantly change. Arrows indicate up- or down-regulated genes in resident macrophages compared to engrafted macrophages at day 12 post transfer. (**B**) Example genes characterizing long-lived intestinal macrophages according to Shaw et al. (**C**) Example genes highly expressed in resident macrophages compared to engrafted cells. (**D**) Example genes highly expressed in engrafted cells compared to long-lived macrophages. (**E**) Example genes participating in IL10/IL10R axis.

The online version of this article includes the following figure supplement(s) for figure 2:

*Figure 2 continued on next page*

*Figure 2 continued*

**Figure supplement 1.** Comparison of graft and engrafted cells to recent monocytic infiltrates in the colon, identified by their Ly6C surface expression.
**Figure supplement 2.** Comparison of colonic macrophages generated upon monocyte transfer into [CD11c-DTR > WT] and [CX3CR1-DTR > WT] BM chimeras treated with DTx.

*2018*). Notably, the majority of the cells we retrieve likely comprise lamina propria/mucosa resident macrophages, rather than the less abundant macrophages of the submucosa or muscularis layer (*Shaw et al., 2018*; *De Schepper et al., 2018*). Generation of a CD4$^+$ Timd4$^+$ subpopulation of lamina propria macrophages was reported to require prolonged residence in the tissue (*Shaw et al., 2018*). Although grafted cells in both colon and ileum acquired with time some expression of a hallmark of the long-lived cells, the phosphatidyl serine receptor Timd4 (Tim4) (*Figure 2B*, *Figure 1D* cluster VI), other signature genes, such as *Cd209f*, *C2* and *Rusc2* (*Shaw et al., 2018*), were not expressed in the time frame analyzed here (*Figure 2B*). The hypothesis that 12 days were insufficient for engrafted macrophages to acquire the full gene expression profile of resident cells was furthermore corroborated by the delayed onset of MHCII (encoded by *H2-ab1*) expression, one of the markers for intestinal macrophage maturation (*Tamoutounour et al., 2012*; *Schridde et al., 2017*) (*Figure 2C*). Engrafted macrophages were also characterized by lack of expression of *Irf1* and the member of the TAM receptor kinase family *Axl* (*Figure 2C*).

Genes, whose expression was low to absent in resident macrophages from both colon and ileum, but prominent in the engrafted cells (*Figure 1D* cluster VII), included the ones encoding ribosome-associated proteins and histones (*Figure 2D*). As previously shown (*Varol et al., 2009*), the reconstitution of emptied intestinal tissue with monocyte-derived cells involves clonal expansion, that is less likely to occur in physiological setting. In line with this notion, out of the 6383 genes significantly differing between engrafted and resident macrophages from either colon or ileum, 897 were annotated with a Gene Ontology (GO) term associated with 'proliferation' and 'cell cycle'; 99 of these DEG were low to absent in resident macrophages from both tissues retrieved from non-engrafted $Cx_3cr1^{GFP/+}$ mice, such as *Hmga1* (*Figure 1D*).

We recently reported the requirement of IL10Ra on colonic macrophages for gut homeostasis. Mice lacking this cytokine receptor on intestinal macrophages develop severe gut inflammation in the colon, but not the ileum (*Zigmond et al., 2014*; *Bernshtein et al., 2019*), as do children that harbor an IL10RA deficiency (*Glocker et al., 2009*). Interestingly, *Il10ra* was not induced in engrafted colonic macrophages, even 12 days after tissue entry, while engrafted ileal macrophages displayed *Il10ra* transcripts as early as day 4 (*Figure 2E*). Expression of the cytokine IL10 itself was almost absent from monocytes and engrafted macrophages in both tissues, but significantly present in resident macrophages retrieved from non-engrafted $Cx_3cr1^{GFP/+}$ mice, to a larger extent in the colon than the ileum (*Figure 2E*). Of note, *Il1b* expression displayed a similar pattern in line with an earlier suggestion that IL1 might induce macrophage IL-10 expression (*Foey et al., 1998*). Likewise, genes induced following macrophage exposure to Il10, such as *Ccr5* (*Houle et al., 1999*) and *Socs3* (*Cassatella et al., 1999*) displayed similar expression patterns. This suggests that in our model the IL10/IL10R axis is inactive in newly differentiated macrophages and established only after further maturation in the tissue.

Flow cytometric analysis of colon tissue of WT C57BL/6 mice identifies a small population of CD11b$^+$ Ly6C$^+$ MHCII$^-$ cells that likely represent recent monocyte immigrants into the tissue (*Figure 2—figure supplement 1A*). These rare cells probably entered the lamina propria to maintain the steady state macrophage pool of the intestine, before differentiating and have been referred to as P1 population of a 'monocyte waterfall' (*Tamoutounour et al., 2012*). Global RNAseq analysis of this population, alongside the Ly6C$^+$ MHCII$^-$ macrophages revealed that, like the day four graft, these endogenous gut immigrants down-regulated monocyte markers and gained a gut macrophage signature, including expression of the Forkhead transcription factor (TF) FoxD2 and the nuclear receptor Nr3c2 (*Figure 2—figure supplement 1B,C*). Endogenous gut immigrants did not display a signature indicating proliferation, such as *Rpl10* or *Hmga1*. However, like the grafted cells, these early colonic immigrants lacked signature genes associated with long-lived gut macrophages, including *Timd4* and *CD209f*, as well as expression of *Il10ra* and *Il10*.

To further corroborate our data, we used a distinct cell ablation model and performed adoptive monocyte transfers into DTx-treated [CX$_3$CR1-DTR > WT] BM chimeras (*Diehl et al., 2013*; *Aychek et al., 2015*). The gene lists of upregulated and down regulated genes in macrophages retrieved from engrafted [CD11c-DTR > WT] and [CX$_3$CR1-DTR > WT] chimeras showed with 71% and 59%, respectively, considerable overlap (*Figure 2—figure supplement 2*). While our adoptive transfer inherently aims at the reconstitution of ablated cells which differ in the two models, the observed coherence suggests robustness of the approach.

Collectively, these data show that despite some differences, monocyte graft-derived cells recapitulate the 'monocyte waterfall' (*Tamoutounour et al., 2012*).

## Monocyte differentiation into gut segment-specific macrophages

We next focused on factors that might be involved in the generation of segment-specific macrophages, that is genes whose expression differed between colonic and ileal macrophages.

458 genes were up-regulated during monocyte differentiation in a segment-specific manner – 351 in colonic macrophages and 107 in ileal macrophages (*Figure 3A*). Monocyte differentiation into ileal macrophages was accompanied by induction of Gata and Hbox TF family members, such as *Gata5* and *Hbox3* (*Figure 3A*), as well as genes encoding the chemokine Ccl5 and the chemokine receptor CCR9. Monocytes that entered colon tissue preferentially up-regulated Foxd2, the nuclear receptor Nr3c2, and the dominant negative helix-loop-helix protein Id2 (*Figure 3A*).

Of the genes, which were specifically down-regulated in only one intestinal segment, 78 genes followed this trend in colonic and 99 in ileal macrophages. The latter down-regulated genes included 7 TFs, such as *Foxp1* and *Trim16*, while *Arid5a* and *Elk3* were specifically down-regulated in the colon (*Figure 3B*).

Many genes related to immune reaction and response to challenge displayed higher expression in ileal macrophages than their colonic counterparts. Examples are: *Arid5a*, whose gene product regulates IL6 (*Masuda et al., 2016*); *Elk3*, which encodes a member of the ETS TF family and was reported to modulate the phagocytosis of bacteria by macrophages (*Tsoyi et al., 2015*) and *Ano6*, that is down-regulated in colonic macrophages (*Figure 3B*), and reportedly supports microbiocidal activity of macrophages involving P2X$_7$ receptor signaling (*Ousingsawat et al., 2015*). In contrast, the enzyme Sod1, which was reported to impair macrophage-related parasite killing in cutaneous Leishmaniasis (*Khouri et al., 2009*), showed lower expression in ileal macrophages (*Figure 3B*).

Another interesting group of genes are those, which are expressed in monocytes, but further up-regulated in one tissue upon differentiation and down-regulated in the other. These genes might encode factors whose expression is incompatible with segment-specific macrophage fates. 54 such genes were expressed in colonic engrafted macrophages and silenced in their ileal counterparts; 15 genes followed an opposite trend (*Figure 3C*). Only one TF was found in both groups, KLF4 (*Figure 3C*). Aldh2 encoded by *Aldh2*, mostly known for its role in alcohol detoxification, was recently reported to play a role in repression of ATP6V0E2, which is critical for proper lysosomal function, autophagy, and degradation of oxidized LDL (*Zhong et al., 2019*) (*Figure 3C*). Hdac7 encoded by *Hdac7* and up-regulated in ileal macrophages and down-regulated in colonic macrophages, was reported to interfere with the myeloid gene expression pattern and to inhibited macrophage-specific functions (*Barneda-Zahonero et al., 2013*) (*Figure 3C*). Finally, *Fmnl3* participates in filopodia generation (*Harris et al., 2010*) (*Figure 3C*).

Collectively, these data establish that monocytes establish gut segment specific gene expression patterns, likely under the influence of local cues.

## Monocyte differentiation into generic tissue macrophages

Transcriptomes of colonic and ileal macrophages 4 days after monocyte tissue entry were alike, with many genes sharing expression in both tissues when compared to their monocyte progenitors. Overall, by day 4, expression of 2007 genes was down-regulated more than 2-fold in the monocyte graft following differentiation into macrophages in both tissues (out of 12485 genes expressed). 2404 genes were induced in both the colon and ileum, arguably as part of a generic transcriptome signature of intestinal macrophages (*Figure 4A*).

Notably, 919 of the genes induced during monocyte differentiation into generic intestinal macrophages displayed very low prior expression in monocytes – below 50 reads (*Figure 4B*). In contrast,

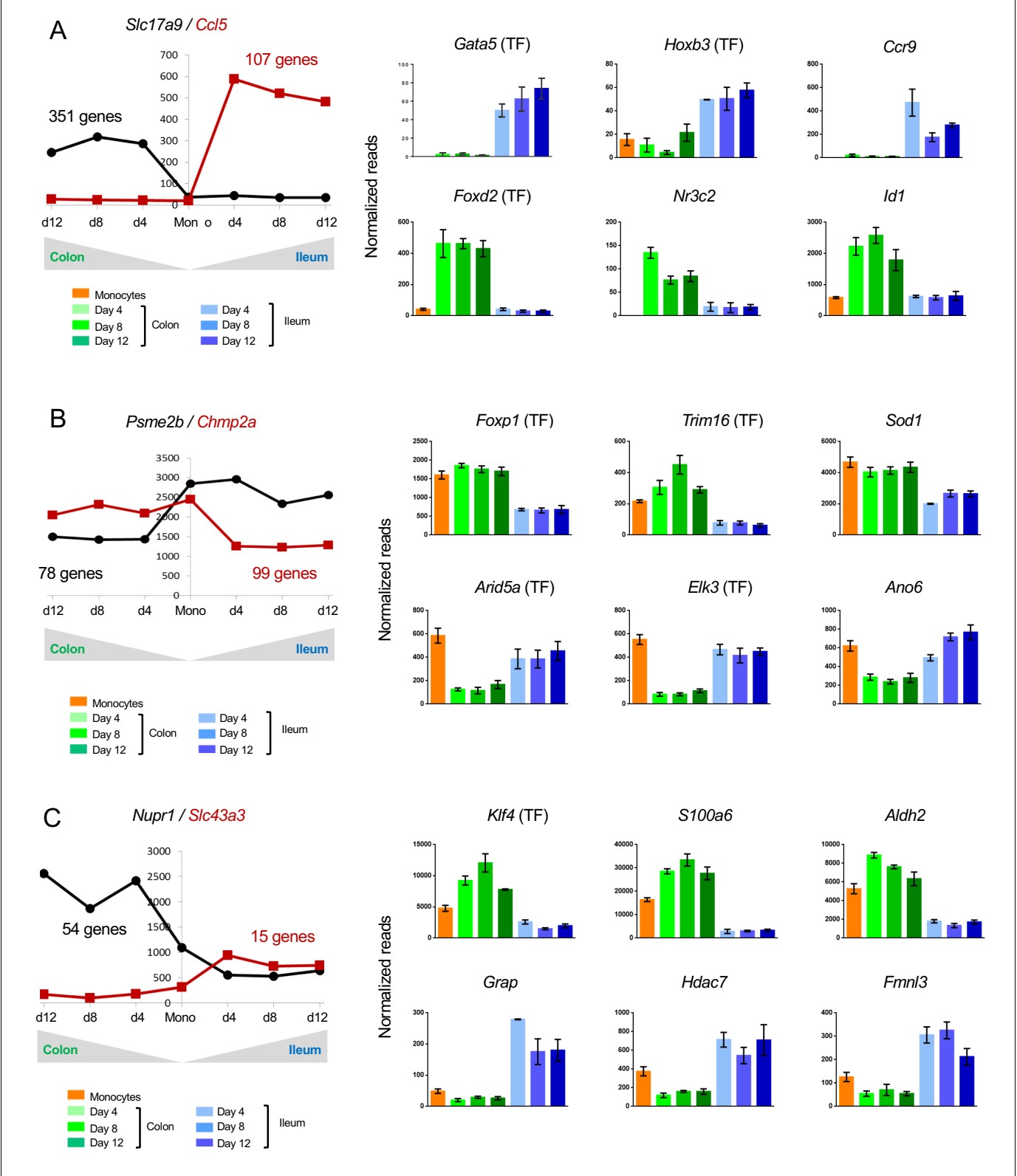

**Figure 3.** Changes in gene expression during conversion of monocytes into colonic or ileal macrophages. (**A**) Example genes that are significantly up-regulated in intestinal macrophages from monocytes to day 4: either up-regulated in the colon but do not change in the ileum, or significantly up-regulated in the ileum but do not change in the colon. Numbers indicate number of genes to follow the trend. Only genes which have significantly different levels between colonic and ileal macrophages at all time points (monocytes > day 4, day4 > day8, day8 > day12) were selected. (**B**) Example
*Figure 3 continued on next page*

*Figure 3 continued*

genes that are significantly down-regulated from monocytes to day four in intestinal macrophages: either down-regulated in the colon but do not change in the ileum, or significantly down-regulated in the ileum but do not change in the colon. Numbers indicate number of genes to follow the trend. (C) Example genes that are either up-regulated in the colon from monocytes to day four and down-regulated in the ileum from monocytes to day four or vice versa.

expression of fewer transcripts (183) seemed to be actively silenced during the differentiation process, as seen in the violin blot in *Figure 4B*. This implies that monocytes actively acquire macrophage identities by de novo mRNA synthesis, while much of the monocytic gene expression is compatible with the differentiation process.

The top 5 GO-terms associated with genes up-regulated in intestinal macrophages related to *cell adhesion and migration* processes (*Figure 4C*), including the chemokine Cxcl1, Ptprk which regulates cell contact and adhesion, and the integrin Itga6 (*Figure 4D*). The top 5 GO-terms associated

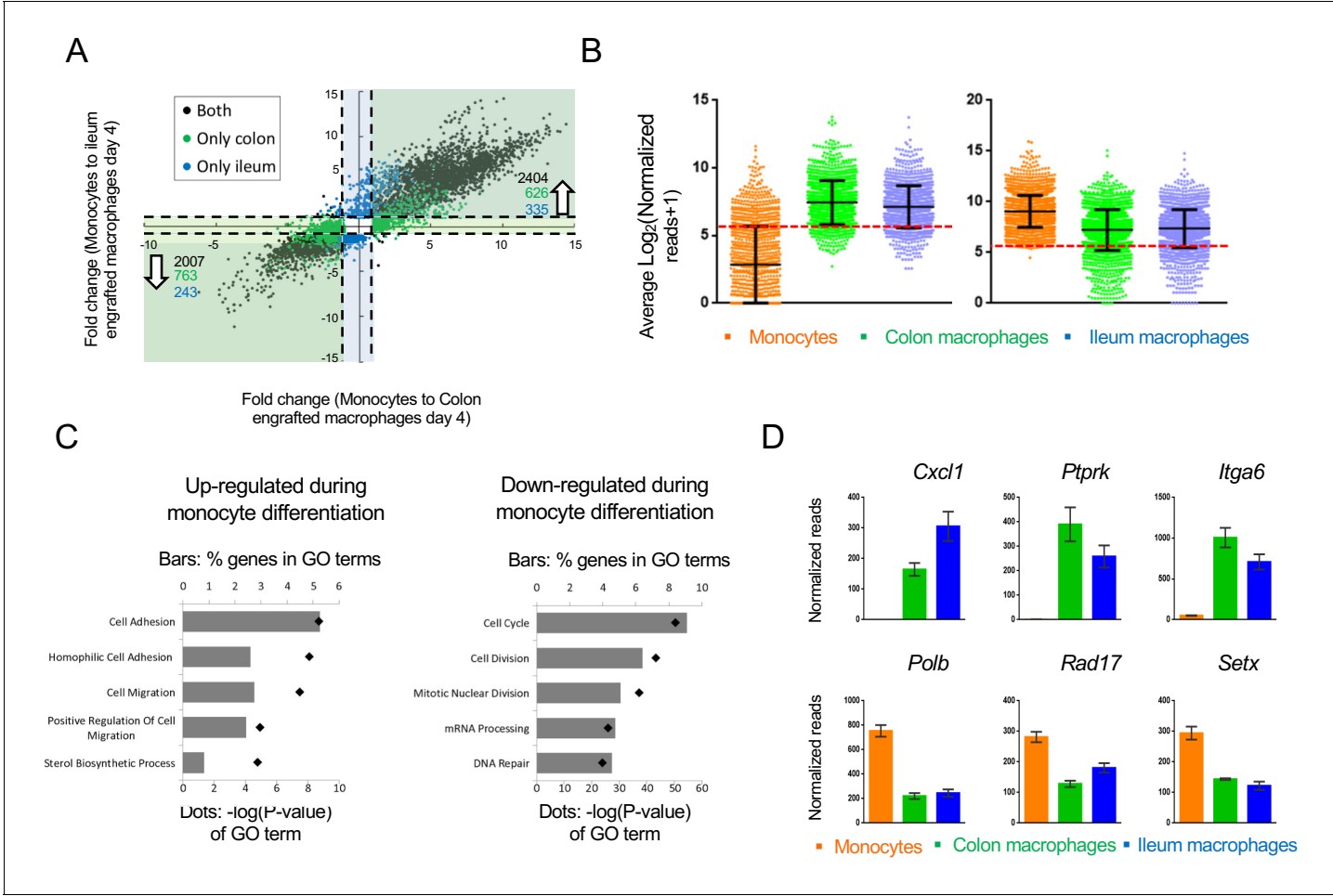

**Figure 4.** Changes in gene expression during conversion of monocytes into generic intestinal macrophages. (A) Dot plot for genes whose expression changes at least 2-fold between both monocytes to engrafted colonic macrophages at day four and monocytes to engrafted ileal macrophages at day 4. Black dots: genes that significantly change in the transition to both tissues. Blue dots: genes that significantly change in monocytes to engrafted ileum macrophages only. Green dots: genes that significantly change from monocytes to engrafted colon macrophages only. Only genes that significantly changed from monocytes to both colonic and ileal macrophages at day 4, but expression levels not distinct between colon and ileum at day 4, were selected. (B) Log averages of all genes that are up-regulated (left) or down-regulated (right) in generic intestinal macrophages compared to monocytes. Red line marks threshold (50 reads, 5.672 in log$_2$) of very low/no expression levels. (C) Top 5 GO pathways that are related to genes that are 'selectively expressed in generic intestinal macrophages compared with monocytes (left) or selectively expressed in monocytes compared with macrophages (right). Bars show percent of genes (out of total genes) that are related to each GO term, red markers illustrate the p-value of each GO term. (D) Example genes from GO pathways. Bars indicate mean normalized reads, error bars represent SEM.

with down-regulated genes in macrophages included *cell cycle and division*, as well as *mRNA processing* and *DNA repair* (**Figure 4C,D**), for example DNA polymerase beta (*Polb*), a cell cycle checkpoint regulator (*Rad17*) and an RNA helicase (*Setx*). Collectively, these data are in line with the transformation of circulating monocytes into non-migratory tissue resident cells and suggest a role for DNA damage-associated molecules during the differentiation process.

Gene expression changes are driven by TFs. In the case of intestinal macrophages, three major TF families seem to participate in monocyte-macrophage differentiation: CCAAT-enhancer-binding proteins (C/EBPs), E2 transcription factors (E2F) and early growth response TFs (Egr). Four of the Cebp family members (*Cebpa, b, d, g*) and 5 E2F family members (*E2f2, 4, 6, 7, 8*) were significantly

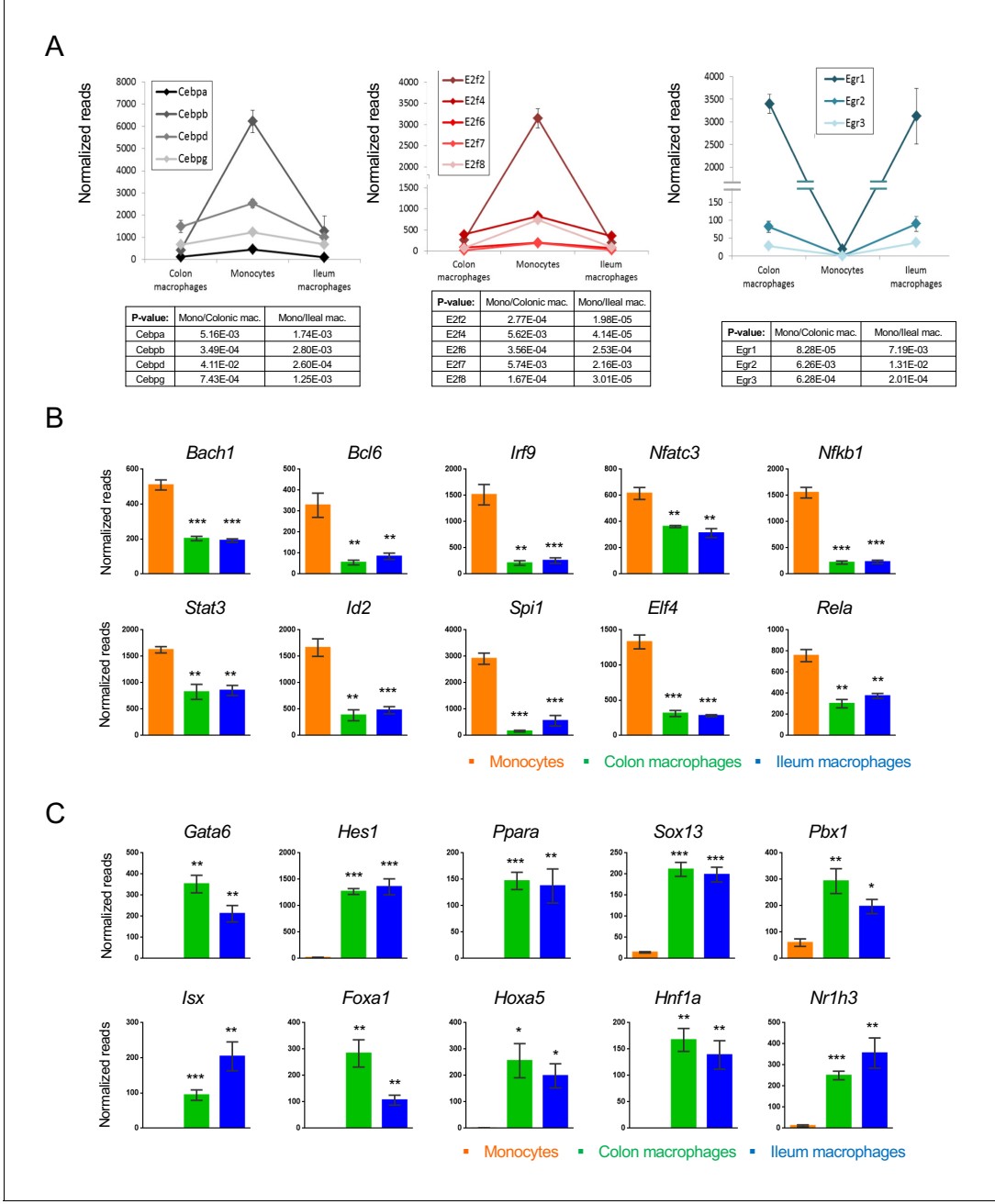

**Figure 5.** Transcription factors which change during intestinal macrophage differentiation. (**A**) Expression graphs of 3 TF families: Cebp (left), E2f (middle) and Egr (right). Lines mark average normalized reads, error bars represent SEM. (**B**) Representative TFs that are down-regulated in generic intestinal macrophages. (**C**) Representative TFs that are up-regulated in generic intestinal macrophages.

down-regulated upon monocyte differentiation into macrophages (*Figure 5A*). In contrast, *Egr1, 2 and 3* were up-regulated. Other down-regulated TFs included the regulators of immune response *Bach1*, *Bcl6*, *Irf9*, *Nfatc3*, *Nfkb1* and *Rela*, as well as the master macrophage TF *Spi1* (PU.1) (*Figure 5B*). The list of induced genes was enriched with homeobox TFs such as *Sox13*, *Pbx1*, *Foxa1* and others (*Figure 5C*). In addition, this group comprised TFs that had previously been reported to be critical for the development of other tissue macrophages, such as the nuclear receptor LXRα encoded by *Nr1h3* for splenic macrophages and Gata6 for peritoneal macrophages (*Varol et al., 2015*; *Rosas et al., 2014*).

## Comparison of tissue-resident and vasculature-resident monocyte-derived cells

Monocytes are generated in the BM to be subsequently disseminated throughout the body via blood vessels. Under inflammation, the cells are rapidly recruited to the site of injury. In absence of challenges, Ly6C$^+$ monocytes can have distinct fates. A fraction of them gives rise to vasculature-resident Ly6C$^-$ 'patrolling' cells (*Auffray et al., 2007*). Other cells contribute to the homeostatic replenishment of selective tissue macrophage compartments (*Figure 6A*). To gauge the impact of the blood environment, as compared to a solid tissue such as the intestine, on the differentiation process, we next compared transcriptomes of Ly6C$^-$ blood cells and gut macrophages to their Ly6C$^+$ monocyte precursors (*Figure 6—figure supplement 1*). A heat map of all 1303 genes, whose expression significantly differed between Ly6C$^+$ monocytes compared to blood- and gut tissue-resident cells (day 4), revealed five clusters (*Figure 6B*). Monocyte progeny, whether in vasculature or tissue shared signatures, showed considerable similarities as reflected in the expression pattern of two thirds of the genes (66.7%). Specifically, clusters I and II comprised 780 genes that were down-regulated upon Ly6C$^+$ monocyte differentiation, including hallmark monocyte markers, such as *Ly6c*, *Ccr2* and *Mmp8* and *Myd88* (*Figure 6B,C*). 160 genes were induced in both blood- and tissue resident monocyte-derived cells, albeit to different extend, including as *Pparg*, *Ets2* and *Tgfbr2* (Cluster III) (*Figure 6B,D*). Cluster IV and V spanned one third of the genes differentially expressed by Ly6C$^+$ monocytes and their progenies, but distinct in tissue and vascular resident cells. Specifically, cluster IV comprised 253 genes up-regulated in blood-resident cells and down-regulated in gut macrophages. This included *Csf2ra*, *Nfkb1*, *Il10ra* and *Spi1* (PU.1). Cluster V comprised 110 genes induced in gut macrophages but not vasculature-resident cells, such as the mitochondrial master regulator *Ppargc1b* and the metalloprotease *Adam19* (*Figure 6B*). Concerning TFs, *Cebpb* was induced in Ly6C$^-$ blood monocytes, as was previously reported (*Mildner et al., 2017*), while *Cebpa* and *Cebpd* were down-regulated in both blood- and gut-resident cells (*Figure 6E*). Irf family members 7, 8 and 9 were down-regulated during Ly6C$^+$ monocyte differentiation in blood and gut.

Collectively, these data establish that vasculature-resident Ly6C$^-$ monocytes and gut macrophages that derive both from Ly6C$^+$ monocytes display considerable overlap in transcriptomic signatures, but also display gene expression patterns that are likely associated with their specific environments.

## Discussion

Adult tissue macrophages can derive from distinct origins (*Ginhoux and Guilliams, 2016*; *Varol et al., 2015*). Most tissue macrophages are currently believed to be generated in the embryo from EMP via a fetal liver monocyte intermediate and subsequently maintain themselves through self-renewal. Selected macrophages residing in barrier tissues, such as gut and skin, however rely on constant replenishment from blood monocytes. Here we report the study of this macrophage generation from monocyte precursors.

Following tissue damage and infection, classical monocytes, defined as CD14$^+$ cells in humans, and Ly6C$^+$ cells in mice critically contribute to inflammatory reactions by promoting and resolving acute challenges (*Ginhoux and Jung, 2014*; *Mildner et al., 2016*). At the sites of injury, monocytes can give rise to cells with both macrophage and DC features. Monocyte differentiation during inflammation has been studied in various pathophysiological settings, including experimental autoimmune encephalitis (*Yamasaki et al., 2014*; *Masuda et al., 2016*), colitis (*Rivollier et al., 2012*; *Zigmond et al., 2012*) and others (*Avraham-Davidi et al., 2013*). To study less well understood physiological monocyte differentiation in absence of overt inflammation, we took advantage of an

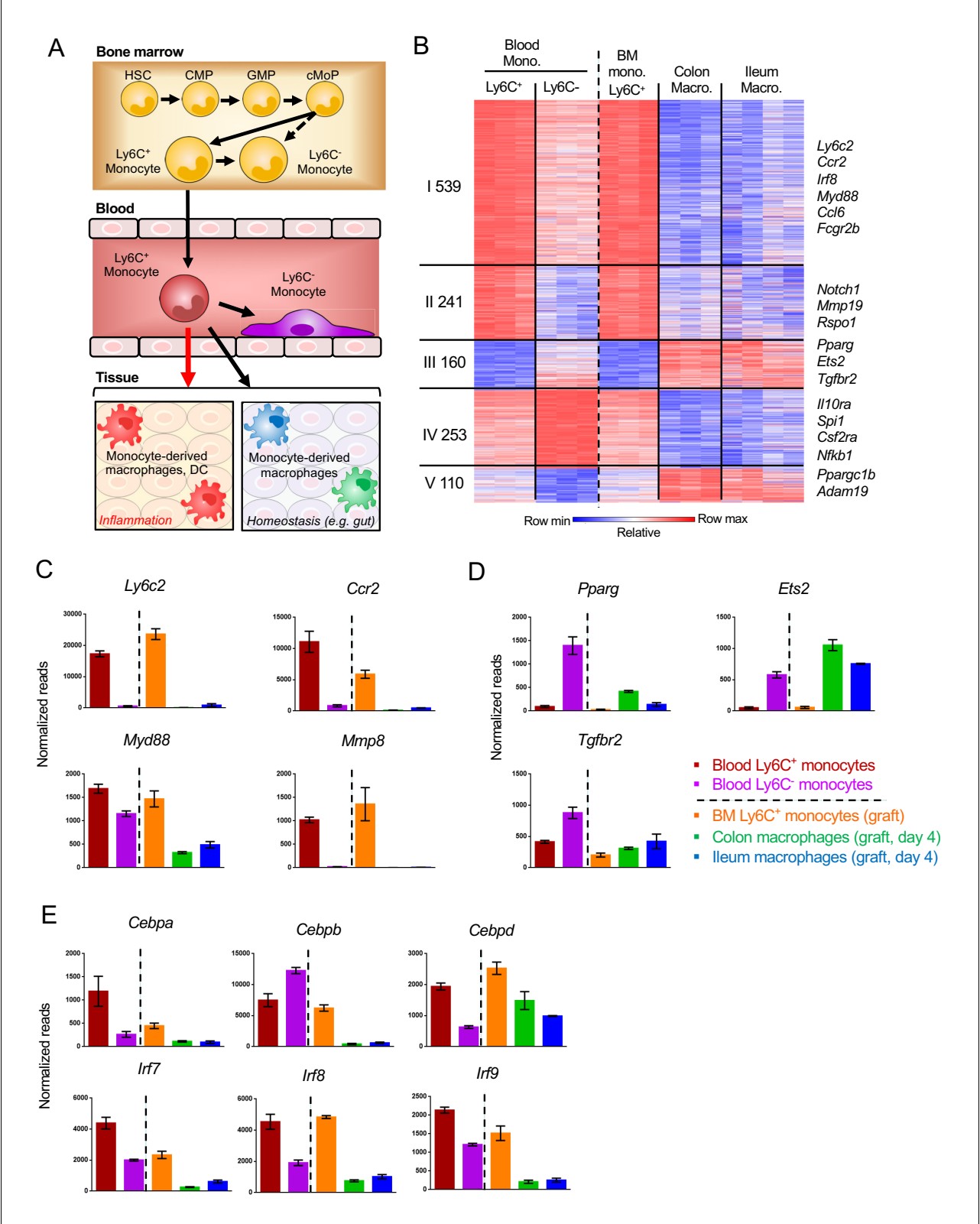

**Figure 6.** Gene expression changes in Ly6C[+] monocytes after differentiating to tissue-resident cells. (**A**) Scheme of monocyte development and fates during physiology and inflammation. (**B**) Fold change heat map of all 1303 genes that significantly change during the transition of Ly6C[+] blood monocytes to Ly6C[-] blood monocytes and BM Ly6C[+] monocytes to engrafted intestinal macrophages at day 4. Colors represent fold change between

*Figure 6 continued on next page*

*Figure 6 continued*

tissue-resident cells and their respective precursor cells. (*Supplementary file 1*, data sets 1, 2, 5, 12,13). (C) Representative monocyte-related genes. (D) Representative genes related to monocyte-derived cells. (E) Expression of TFs of the *Cebp* and *Irf* gene families.

The online version of this article includes the following figure supplement(s) for figure 6:

**Figure supplement 1.** Gating strategy for sorting of Ly6C$^+$ and Ly6C$^-$ blood monocytes.

experimental system that allows synchronized reconstitution of macrophage compartments by monocyte engraftment (*Varol et al., 2007*; *Varol et al., 2009*).

The majority of changes in gene expression, both in fold change and numbers, occurred in the transition from monocytes to either colonic or ileal macrophages at day four post transfer, that is immediately after extravasation. Engrafted colonic and ileal macrophages segregated in gene expression according to tissue. Tissue-specific macrophage imprinting occurs hence early during development, shortly after tissue infiltration. Interestingly, two samples of engrafted ileal macrophages clustered with the colonic samples rather than the ileal ones, suggesting that the colonic signature is default. Colonic macrophages at day 12 post engraftment were also more distinct from endogenous colonic macrophages, compared to their ileal counterparts, which implies that the mature colonic gene signature might take longer time to develop or could be more heterogeneous. Recent studies have noted heterogeneity within murine blood monocytes, in particular with respect to an intermediate between the Ly6C$^+$ and Ly6$^-$ populations (*Mildner et al., 2017*). While we currently cannot formally rule out that, colonic and ileal macrophages could hence derive from distinct precursors, we consider this however unlikely.

Mowat and colleagues have reported a transcriptome analysis of monocytes and colonic macrophages, including intermediates of the 'waterfall' (*Schridde et al., 2017*). The authors highlighted the critical role of TGFb in the differentiation process. While a comparison of these data to ours confirmed the late onset of genes that characterize long-lived gut macrophages, the use of the distinct platforms and distinct experimental set up precluded further direct alignment. Of note, monocyte-derived cells are in our system synchronized with respect to development and therefore allow additional temporal resolution, especially with respect to final population of the scheme (P4), which comprises in the cited study (*Schridde et al., 2017*) a heterogeneous conglomerate.

With their extended half-life, Ly6C$^-$ monocytes that patrol the endothelium, have been proposed to represent vasculature-resident macrophages (*Ginhoux and Jung, 2014*). Indeed, these cells shared gene signatures with the gut resident macrophages, such as the reduction in IRF TFs following monocyte differentiation and induction of characteristic macrophage genes, such as PPARg and TGFbR. However, the comparison of these blood-resident cells to gut tissue-resident macrophages revealed also considerable differences likely associated with the residence in vasculature and the solid tissue, respectively.

Though anatomically close, the small and large intestine represent very distinct tissues, including structural dissimilarities, such as the extended ileal villi and Peyer's Patches, characteristic distinct abundance of immune cells, as well as different luminal microbiome content (*Mowat and Agace, 2014*). Highlighting these differences, ileum and colon display also unique susceptibility to perturbations, as for instance to a IL10R deficiency (*Zigmond et al., 2014*; *Bernshtein et al., 2019*). Our comparative analysis of colonic and ileal macrophages, including their generation from monocytes, might provide critical insights into the mechanism underlying segment-specific pathology resistance or - susceptibility in the gut. Together with earlier reports (*Mildner et al., 2017*; *Schridde et al., 2017*), our data sets can provide a starting point for hypothesis-driven experiments.

To conclude, we characterized here monocyte-derived intestinal macrophages generated under conditions avoiding overt inflammation. We highlight specific genes and TFs which are regulated following monocyte differentiation to generic or segment-specific intestinal macrophages. By comparing transcriptomes of early intestinal macrophages and blood-resident Ly6C$^-$ cells, we show that the populations which share a common ancestor – the Ly6C$^+$ blood monocytes – show considerable overlap in gene expression, while they also display adaptation to their specific environments. Our data provide a gateway and reference point to further studies on monocyte differentiation to macrophages.

## Materials and methods

### Mice

Mice were kept in a specific-pathogen-free (SPF), temperature-controlled (22 ± 1°C) facility on a reverse 12 hr light/dark cycle at the Weizmann Institute of Science. Food and water were given ad libitum. Mice were fed regular chow diet (Harlan Biotech Israel Ltd, Rehovot, Israel). The following mice strains all on C57BL6 background were used: *Cx3cr1$^{gfp/+}$* mice (*Jung et al., 2000*), CD11c-DTR transgenic mice (B6.FVB-Tg [Itgax-DTR/GFP] 57Lan/J) (*Jung et al., 2002*) and CX3CR1-DTR transgenic mice (*Diehl et al., 2013*). BM chimeras were generated by engraftment of 7–10 weeks old recipient mice that were irradiated the day before with a single dose of 950 cGy using a XRAD 320 machine (Precision X-Ray (PXI). Femurs and tibiae of donor mice were removed and BM was flushed with cold PBS. BM was washed with cold PBS twice and filtered by 100 µm filter. BM cells were suspended in PBS and $5 \times 10^6$ cells were injected IV into irradiated recipient. Mice were handled and experiments were performed under protocols approved by the Weizmann Institute Animal Care Committee (IACUC) in accordance with international guidelines.

### Isolation of BM monocyte grafts and monocyte transfers

Femurs and tibias of donor mice were removed and BM was flushed with cold PBS. BM was washed with cold PBS twice and filtered by 100 µm filter. Cells were suspended with PBS and loaded on equal amount of Ficoll (GE healthcare). Tubes were centrifuged 920 g in room temperature for 20 min without breaks and Buffy coats were collected and washed with cold PBS. CD11c-DTR > wt]. Cells were stained and sorted according to the following markers: CD117$^-$ CD11b$^+$ CD115$^+$ Ly6C$^+$ GFP$^{int}$. BM chimeras were treated with 18 ng / gram bodyweight Diphtheria toxin (DTx) (Sigma-Aldrich, Cat # D0564) for two consecutive days before transfer. At the day of transfer mice were injected with $10^6$ BM monocytes IV. At days 1, 3, 5, 7 and 9 after transfer mice were injected with 9 ng / gr bodyweight DTx.

### Isolation of intestinal lamina propria cells

Intestines were removed and fecal content flushed out with PBS; tubes were opened longitudinally and cut into 0.5 cm sections. Pieces were placed in 5ml/sample (up to 300gr of tissue) of Hanks' Balanced Salt Solution (HBSS) with 10% heat-inactivated FCS/FBS, 2.5 mM EDTA and 1 mM DL-Dithiothreitol ((DTT), Sigma-Aldrich Cat# D9779) and incubated on a 37°C shaker at 300 rpm for 30 min to remove mucus and epithelial cells. Following incubation, samples were vortexed for 10 s and filtered through a crude cell strainer. Pieces that did not pass the strainer were collected and transferred to 5 ml/sample of PBS +/+ with 5% heat-inactivated FCS/FBS, 1 mg/ml Collagenase VIII (Sigma-Aldrich Cat# C2139) and 0.1 mg/ml DNase I (Roche Cat# 10104159001). Tissue was incubated in a 37°C shaker at 300 rpm for 40 min (colon) or 20 min (ileum) in the solution. After incubation, samples were vortexed for 30 s until tissue was dissolved, then filtered through a crude cell strainer. The strainer was washed with PBS - /- and centrifuged in 4°C, 375G for 6 min. Cells were stained and subjected to FACS analysis or sorting.

### Isolation of murine blood monocytes

Blood was retrieved from the vena cava, immediately placed in 150 U/ml Heparin and loaded on Ficoll (GE healthcare). Tubes were centrifuged 920 g in room temperature for 20 min without breaks and Buffy coats were collected and washed with cold PBS. Cells were sorted according to the following parameters: CD45$^+$ CD11b$^+$ CD115$^+$ Ly6C$^{+/-}$.

### Flow cytometry analysis

Samples were suspended and incubated in staining medium (PBS without calcium and magnesium with 2% heat-inactivated Fetal Calf/Bovine Serum (FCS/FBS) and 1 mM EDTA) containing fluorescent antibodies. Following incubation, cells were washed with staining buffer only or staining buffer with DAPI, centrifuged, filtered through 80 µm filter and read. For FACS analysis, LSRFortessa (BD Biosciences) was used. For cell sorting, FACSAria III or FACSAria Fusion (BD Biosciences) were used. Results were analyzed in FlowJo software (Tree Star). Staining antibodies (clones indicated within

brackets): anti-CD45 (30-F11), CD11b (M1/70), CD115/CSF-1R (AF598), Ly-6C (HK1.4), CD64/FcγRI (X54-5/7.1), CD11c (N418), anti-I-A$^b$ (MHCII) (AF6-120.1), DAPI.

## RNA sequencing and analysis

RNA-seq of populations was performed as described previously (*Diehl et al., 2013*; *Jaitin et al., 2014*). Cells were sorted into 100 µl of lysis/binding buffer (Life Technologies) and stored at 80°C. mRNA was captured using Dynabeads oligo(dT) (Life Technologies) according to manufacturer's guidelines. A derivation of MARS-seq (*Jaitin et al., 2014*) was used to prepare libraries for RNA-seq, as detailed in *Shemer et al. (2018)*. RNA-seq libraries were sequenced using the Illumina Next-Seq 500. Raw reads were mapped to the genome (NCBI37/mm9) using hisat (version 0.1.6). Only reads with unique mapping were considered for further analysis. Gene expression levels were calculated and normalized using the HOMER software package (analyzeRepeats.pl rna mm9 -d < tagDir > count exons -condenseGenes -strand + -raw). Gene expression matrix was clustered using k-means algorithm (MATLAB function kmeans) with correlation as the distance metric. PCA was performed by MATLAB function pca. Gene ontology was performed by DAVID (https://david.ncifcrf.gov). Data on molecules and pathways was partly obtained by Ingenuity Pathway Analysis (IPA), Ingenuity Target Explorer, Qiagen and Metascape Pathway analysis (*Zhou et al., 2019*).

## Statistics

Results are presented as mean ± SEM. Statistical analysis was performed using Student's t test. * p-value<0.05 ** p-value<0.01 *** p-value<0.001.

## Acknowledgements

The authors declare no competing financial interests. We would like to thank all members of the Jung laboratory for helpful discussion. We further thank the staff of the Weizmann Animal facility and the members of the FACS facility for expert advice.

## Additional information

### Funding

| Funder | Grant reference number | Author |
| --- | --- | --- |
| European Research Council | 340345 | Steffen Jung |

The funders had no role in study design, data collection and interpretation, or the decision to submit the work for publication.

### Author contributions

Mor Gross-Vered, Conceptualization, Data curation, Investigation, Methodology, Writing—original draft, Writing—review and editing; Sébastien Trzebanski, Validation, Investigation, Methodology; Anat Shemer, Formal analysis, Investigation; Biana Bernshtein, Data curation, Investigation, Writing—review and editing; Caterina Curato, Formal analysis, Methodology, Writing—review and editing; Gil Stelzer, Data curation; Tomer-Meir Salame, Methodology; Eyal David, Data curation, Software, Methodology; Sigalit Boura-Halfon, Supervision, Methodology, Writing—review and editing; Louise Chappell-Maor, Resources, Data curation, Methodology; Dena Leshkowitz, Data curation, Software; Steffen Jung, Conceptualization, Supervision, Funding acquisition, Writing—original draft, Project administration, Writing—review and editing

### Author ORCIDs

Sébastien Trzebanski (iD) http://orcid.org/0000-0002-0647-6814
Caterina Curato (iD) http://orcid.org/0000-0003-4938-7288
Steffen Jung (iD) https://orcid.org/0000-0003-4290-5716

### Ethics

Animal experimentation: Mice were handled and experiments were performed under protocols approved by the Weizmann Institute Animal Care Committee (IACUC) in accordance with international guidelines.

### Decision letter and Author response

Decision letter https://doi.org/10.7554/eLife.49998.sa1
Author response https://doi.org/10.7554/eLife.49998.sa2

## Additional files

### Supplementary files

• Source data 1. Excel sheets of the data which formed the basis of *Figures 1–4,* and *6*.

• Supplementary file 1. List of RNAseq data sets prepared and used in this study.

• Transparent reporting form

### Data availability

Data have been deposited on GEO under the accession number GSE140919. A source data file has been provided.

The following dataset was generated:

| Author(s) | Year | Dataset title | Dataset URL | Database and Identifier |
|---|---|---|---|---|
| Gross-Vered M, Trzebanski S, She-mer A, Bernshtein B, Curato C, Stelzer G, Salame T, David E, Boura-Halfon S, Chappell-Maor L, Dena Leshkowitz, Steffen Jung | 2020 | Defining murine monocyte differentiation into colonic and ileal macrophages | https://www.ncbi.nlm.nih.gov/geo/query/acc.cgi?acc=GSE140919 | NCBI Gene Expression Omnibus, GSE140919 |

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
