## [Decision Letter]

**Acceptance summary:**

The paper shows transcriptomic studies of diphtheria toxin ablation of CD11c cells with engraftment of CX3CR1 donor cells in a well-established macrophage reconstitution model. The main conclusions of the paper are that most transcriptomic signatures of the intestinal context (compared with blood monocytes) are rapidly acquired; that there are distinct transcriptomic signatures between the small and large intestinal macrophages and that early uptake differences in line with the Mowat 'waterfall' effect can be captured.

This is a well-established model by a group with a strong track-record in the field addressing an important problem to understand the differences in intestinal macrophages between the small and large intestines.

**Decision letter after peer review:**

[Editors’ note: the authors were asked to provide a plan for revisions before the editors issued a final decision, this was subsequently approved by the editors. What follows is the editors’ letter requesting such plan.]

Thank you for sending your article entitled "Defining monocyte differentiation into colonic and ileal macrophages" for peer review at *eLife*. Your article is being evaluated by three peer reviewers, one of whom is a member of our Board of Reviewing Editors, and the evaluation is being overseen by Satyajit Rath as the Senior Editor.

Given the list of essential revisions, including new experiments, the editors and reviewers invite you to respond with an action plan and timetable for the completion of the additional work. We plan to share your responses with the reviewers and then issue a binding recommendation.

There are two main issues where we would be very grateful for clarification. The first is how you would intend to respond to the potential bias introduced by the CD11c depletion strategy as in the major comment of reviewer 2 and the general remarks of reviewer 3. A second vital issue is that the paper could easily be very quickly superseded by a single cell approach. If you have, or intend to obtain single cell data, we would ask that it is included in this paper. Of course, we would respect your wishes if you choose to publish this data and the single cell data elsewhere, although we would have concerns if the *eLife* manuscript is preliminary to another publication.

Reviewer #1:

1) A concern is that the specifics of the k-means clustering with subsequent selectivity of individual transcripts. Justification of the groupings (elbow, gap statistics) is not shown, and it is generally unclear how multiplicity of testing is dealt with. It would likely transform the paper to show 'pseudo-time' analyses (E.g. Nature Communications 9:2442). In addition, color-coding the bulk plots for the different cell populations would give a feeling for the groupings independently of the most extreme cases. It would be helpful to rationalize results according to discrimination of different intestinal contexts and may be more powerful in revealing contextual differences.

2) The authors (almost certainly correctly) argue that lamina propria macrophages are the main population being studied. Nevertheless, they are showing bulk analyses of heterogeneous populations (developmentally and according to micro-anatomical position). This data seems extremely likely to be rapidly superseded by single cell analyses. At the very least the authors should acknowledge this point. It would rather devalue the impact of the paper in *eLife* if the authors are able to publish single cell analyses in a different paper in the near future: to include such data would empower the above approach and transform the paper.

3) It appears that many of the annotated gene expression results selected for individual display in the histograms are derived from the bulk transcriptomic analyses. The question of reproducibility could be dealt with by qPCR methodology for critical targets.

4) The manuscript rather catalogues the microphage contextual differences without clear validation. It would help to use in enrichment plots or STRING analyses to display the biological interpretations.

Reviewer #2:

The present manuscript by Steffen Jung and collaborators entitled "Defining monocyte differentiation into colonic and ileal macrophages" represents a well-written report by experienced leaders in the field of macrophage/dendritic cell biology.

The authors used CD11c-DTR bone marrow chimeric mice to deplete CD11c+ macrophages before reconstituting with CD45.1 congenic Ly6C^+^ bone marrow-derived monocytes. This allowed to synchronize, and, comparatively analyze, the differentiation of monocyte-macrophage differentiation under homeostatic conditions during a 12 day period in the ileum, and the colon by RNA profiling.

Hence, the authors addressed a so far under-investigated topic, i.e. the site-specific differentiation/maturation of monocyte-derived macrophages in distinct intestinal compartments under non-inflammatory, homeostatic conditions.

Novel findings reported include the identification of distinct expression profiles of macrophages in separate intestinal segments under homeostatic conditions, likely driven by site-specific microenvironmental cues which (likely) lead to the distinct expression profiles seen in resident macrophages in the ileum and the colon. Remarkably, following engraftment, the monocyte-intestinal macrophage differentiation essentially recapitulated the monocyte-macrophage waterfall as previously described by Bernard Malissen and colleagues.

Last but not least, I also agree with their conclusion that "their data provide a gateway and reference point to further studies on monocyte differentiation to macrophages" – despite some discrepant findings to reports in the literature -which were also acknowledged by the authors and ascribed to different experimental set-ups and use of distinct analysis platforms.

My main concern with the present manuscript is the used the cell-depletion protocol, which most likely results in a biased depletion of some, but not all, host-derived intestinal macrophage subsets in recipient mice:

The authors took advantage of a (CD11c-DTR)-targeted cell-depletion-reconstitution strategy to obtain a synchronized reconstitution of the intestinal macrophage compartment by the adoptive transfer of bone marrow-derived Ly6C^+^ monocytes. This experimental system was previously used by the authors, for example, to successfully dissect the different origins and functions of intestinal lamina propria DC subsets (Varol et al., 2009). In this article, the authors also acknowledge that "DTx treatment of [CD11c-DTR > WT] chimeras resulted in the depletion of all lamina propria DC subsets, whereas the population of CD11c-/lo CD11b^+^ lpMø remained unaltered". Hence, in the present study even after repeated DT treatment, the intestinal lamina propria is likely to be still populated with host (CD45.2)-derived intestinal CD11b^+^ CD11c-/lo macrophages at the time of engraftment with the CD45.1 Ly6C^+^ BM-derived monocytes. Hence, it cannot be ruled out that these host-derived macrophages affect the differentiation pathway of the engrafted BM-monocytes in the present study, e.g. by competing with engrafted monocytes for appropriate niches for the further differentiation/maturation in the lamina propria, and/or by further influencing the site-specific microenvironment (e.g. by selective secretion of mediators that affects the differentiation of engrafted monocytes).

Hence, the presence of a biased population of host-derived macrophages during the synchronized differentiation of monocytes to macrophages need to be considered and the possible implications on the interpretations of the results obtained need to be addressed by the authors. As an example, providing information on the relative distribution and frequencies of these CD45.2^+^ CD11c -/lo CD11b^+^ lamina propria macrophages in untreated, and engrafted chimeric CD45.1 mice may allow assessing the consequences of the biased depletion of intestinal macrophages using this particular experimental cell-depletion system.

Reviewer #3:

Gross-Vered et al. use a strategy where they ablate all CD11c+ cells and then transfer blood monocytes to repopulate intestinal macrophages. They then seek to define the transcriptional program of differentiation of monocytes into tissue resident macrophages. At 4 days post-transfer (earliest timepoint looked at), the authors find that cells which have extravasated into the tissue no longer transcriptionally resemble monocytes and are more similar to tissue resident macrophages. The authors also identify some changes between small and large intestine. However, resident cell populations examined are likely a mixed population. Others have shown that monocytes do not reconstitute all intestinal macrophage populations so differences with resident cells could be due to altered cell populations. As the ablation strategy depletes all CD11c+ cells (all lamina propria macrophages and all dendritic cells) it is unclear how physiologically relevant this system is. Finally, functional predictions from the data are lacking, making it difficult to assess significance.

1) What is the resident population? If defined as in Figure 1—figure supplement 1A they are CD11b^+^CD64^+^MHCII^+^ Recent publications (Shaw et al., 2018, Schepper et al., 2018) demonstrated that this contains at least 3 macrophage populations: a self-renewing population, one with slow turnover from monocytes and one with rapid turnover from monocytes. The authors should compare with each population that repopulates (slowly and quickly) from monocytes.

2) The ablation strategy utilized will get rid of all macrophage populations as well as dendritic cells. It is unclear if transferred monocytes can repopulate all macrophage populations. Altered transcriptional profiles could be due to changed proportion of differentiated cells. Similarly, different representation of each population could lead to differences between small and large intestine.

3) Shaw et al., 2018, Schepper et al., 2018, and Schridde et al., 2017, performed transcriptional profiling of these populations during distinct developmental windows or during repopulation after ablation of individual populations. It is unclear how the data presented in Gross-Vered fits with the findings in these publications. In Discussion the authors note minimal overlap with Schridde and claim that differences in finding (which are not elaborated) are because Schridde et al. did not synchronize precursors, but this could be because the ablation strategy removes and does not repopulate a broad range of cell populations.

4) A mapping or network analysis would be helpful to understand how these cell populations could be functionally distinct. It is unclear how any described differences would lead to differential outcomes in the tissue. The authors have superficial analysis with listing of GO terms, but these are quite generic.

5) Unclear what the comparison with monocytes adds. These are functionally very different cells so it is not surprising they have distinct transcriptional profile from tissue resident macrophages.

6) What do the authors mean that since two of the ileal samples cluster with the colon it means the colon is the default? More data would need to support this statement.

---

## [Author Response]

[Editors’ notes: the authors’ response after being formally invited to submit a revised submission follows.]

Reviewer #1:[…] 1) A concern is that the specifics of the k-means clustering with subsequent selectivity of individual transcripts. Justification of the groupings (elbow, gap statistics) is not shown, and it is generally unclear how multiplicity of testing is dealt with. It would likely transform the paper to show 'pseudo-time' analyses (E.g. Nature Communications 9:2442). In addition, color-coding the bulk plots for the different cell populations would give a feeling for the groupings independently of the most extreme cases. It would be helpful to rationalize results according to discrimination of different intestinal contexts and may be more powerful in revealing contextual differences.

We thank the reviewer for these suggestions; In the revised manuscript we now show the elbow plot that was the basis for the heatmap in Figure 1 as Figure 1—figure supplement 3A. We also have explored additional avenues for the analysis of the data with our bioinformaticians and added a metascape pathway analysis to the revised manuscript (Figure 1—figure supplement 3B-E).

2) The authors (almost certainly correctly) argue that lamina propria macrophages are the main population being studied. Nevertheless, they are showing bulk analyses of heterogeneous populations (developmentally and according to micro-anatomical position). This data seems extremely likely to be rapidly superseded by single cell analyses. At the very least the authors should acknowledge this point. It would rather devalue the impact of the paper in eLife if the authors are able to publish single cell analyses in a different paper in the near future: to include such data would empower the above approach and transform the paper.

We understand the concern of the reviewer, but we do not think that our data will be superseded by a sc analysis. Rather this independent approach will remain an important complement. Single cell transcriptome analysis bears its own shortcomings, including the limited transcriptome coverage and the fact that at least droplet-based methods inherently tend to miss certain fragile populations that fail to survive the sorting. Moreover, inference on developmental pathways have with sc transcriptomics to rely on bioinformatic ‘pseudotime analysis’. These approaches can be powerful, as for instance in the elegant RNA velocity method (La Manno, G. et al. Nature 2018), which is based on the distinction of unspliced and spliced mRNAs. However, they clearly require independent confirmation. In contrast, we synchronize the cell development by the ablation/ precursor transfer strategy and thus rely on real time analysis. We believe this remains a critical complement to the sc analyses and will be happy to discuss this further in our manuscript. We however agree that also our approach has its pitfalls, which we openly discussed in our manuscript and will expand on, in response to the constructive reviewer comments we obtained.

For the above reasons, we ourselves decided not to perform a complementary single cell transcriptome analysis on gut macrophages, but rather to launch a mutagenesis approach to functionally test a number of candidate genes that came up in our study. This will however take considerable time and is, we believe, beyond the scope of the current report. At this stage, we however would like to share our findings and data sets with the community.

3) It appears that many of the annotated gene expression results selected for individual display in the histograms are derived from the bulk transcriptomic analyses. The question of reproducibility could be dealt with by qPCR methodology for critical targets.

We would like to point out that all our results are based on repeated global RNAseq analyses and believe reproducibility is therefore ensured. To our mind, qPCR-based additional repeats of selected genes will here not add much.

4) The manuscript rather catalogues the microphage contextual differences without clear validation. It would help to use in enrichment plots or STRING analyses to display the biological interpretations.

We thank the reviewer for this suggestion and have added to the revised manuscript a Metascape analysis (Zhou et al., 2019), as Figure 1—figure supplement 3B-E. This highlights the potential functional differences of gene lists co-expressed by the ileal and colonic macrophages and differentially expressed by the segment specific macrophages.

Reviewer #2:[…] My main concern with the present manuscript is the used the cell-depletion protocol, which most likely results in a biased depletion of some, but not all, host-derived intestinal macrophage subsets in recipient mice:The authors took advantage of a (CD11c-DTR)-targeted cell-depletion-reconstitution strategy to obtain a synchronized reconstitution of the intestinal macrophage compartment by the adoptive transfer of bone marrow-derived Ly6C^+^ monocytes. This experimental system was previously used by the authors, for example, to successfully dissect the different origins and functions of intestinal lamina propria DC subsets (Varol et al., 2009). In this article, the authors also acknowledge that "DTx treatment of [CD11c-DTR > WT] chimeras resulted in the depletion of all lamina propria DC subsets, whereas the population of CD11c-/lo CD11b^+^ lpMø remained unaltered". Hence, in the present study even after repeated DT treatment, the intestinal lamina propria is likely to be still populated with host (CD45.2)-derived intestinal CD11b^+^ CD11c-/lo macrophages at the time of engraftment with the CD45.1 Ly6C^+^ BM-derived monocytes. Hence, it cannot be ruled out that these host-derived macrophages affect the differentiation pathway of the engrafted BM-monocytes in the present study, e.g. by competing with engrafted monocytes for appropriate niches for the further differentiation/maturation in the lamina propria, and/or by further influencing the site-specific microenvironment (e.g. by selective secretion of mediators that affects the differentiation of engrafted monocytes).Hence, the presence of a biased population of host-derived macrophages during the synchronized differentiation of monocytes to macrophages need to be considered and the possible implications on the interpretations of the results obtained need to be addressed by the authors. As an example, providing information on the relative distribution and frequencies of these CD45.2^+^ CD11c -/lo CD11b^+^ lamina propria macrophages in untreated, and engrafted chimeric CD45.1 mice may allow assessing the consequences of the biased depletion of intestinal macrophages using this particular experimental cell-depletion system.

We agree with the reviewer that our approach that is based on a combination of cell ablation and adoptive precursor transfer has like any other model limitations, and have discussed these in our manuscript.

The novel messages of our study are that (1) intestinal macrophages gradually acquire their characteristic transcriptome signature and (2) ileal and colonic macrophages gradually acquire distinct signatures.

We assume that in colon and ileum the same cells are ablated and reconstituted. Cells that remain, or cells that are generated from endogenous monocytes will likely be the same in the two tissues.

We also would like to draw attention to Figure 1—figure supplement 2B. There we show that endogenous CD64^+^ cells are as expected depleted of CD11c^hi^ cells. Remaining CD11c^lo^ cells all still express Ly6C indicating their recent derivation from monocytes. In contrast, the engrafted CD45.1 cells display the normal profile of gut macrophages (with CD11c^hi^ abundance > CD11c^lo^ abundance) and essential absence of Ly6C.

We consider an impact of CD11c^lo^ endogenous macrophages on the engrafted cells, as suggested by the reviewer, rather unlikely. In the CD11c-DTR system we deplete CX_3_CR1^+^ macrophages most of which express CD11c. Of note, we have obtained similar results with monocyte transfers in a [CX3CR1-DTR > WT] system where all CX3CR1-expressing macrophages are deleted, and include these in the revised manuscript as Figure 2—figure supplement 6. We also included a statement that our analysis refers in both tissues only to the population of replaced cells.

Reviewer #3:Gross-Vered et al. use a strategy where they ablate all CD11c+ cells and then transfer blood monocytes to repopulate intestinal macrophages. They then seek to define the transcriptional program of differentiation of monocytes into tissue resident macrophages. At 4 days post-transfer (earliest timepoint looked at), the authors find that cells which have extravasated into the tissue no longer transcriptionally resemble monocytes and are more similar to tissue resident macrophages. The authors also identify some changes between small and large intestine. However, resident cell populations examined are likely a mixed population. Others have shown that monocytes do not reconstitute all intestinal macrophage populations so differences with resident cells could be due to altered cell populations. As the ablation strategy depletes all CD11c+ cells (all lamina propria macrophages and all dendritic cells) it is unclear how physiologically relevant this system is. Finally, functional predictions from the data are lacking, making it difficult to assess significance.

We agree that the functional impact of the differential transcriptomes is what eventually counts. Bioinformatic network analysis can further help to make prediction but arguably will remain hand waving. We therefore have launched a mutagenesis approach to functionally test a number of candidate genes that came up in our study. This will however take considerable time and is, we believe, beyond the scope of the current report. At this stage, we however would like to share our findings and data sets with the community.

1) What is the resident population? If defined as in Figure 1—figure supplement 1A they are CD11b^+^CD64^+^MHCII^+^ Recent publications (Shaw et al., 2018, Schepper et al., 2018) demonstrated that this contains at least 3 macrophage populations: a self-renewing population, one with slow turnover from monocytes and one with rapid turnover from monocytes. The authors should compare with each population that repopulates (slowly and quickly) from monocytes.

The novel messages of our study are that:

1) intestinal macrophages gradually acquire their characteristic transcriptome signature and;

2) ileal and colonic macrophages gradually acquire distinct signatures.

Comparison to the resident population serves as a reference point. As also argued below, in our hands the use of distinct platforms and experimental set ups renders alignment of data from different studies problematic.

2) The ablation strategy utilized will get rid of all macrophage populations as well as dendritic cells. It is unclear if transferred monocytes can repopulate all macrophage populations. Altered transcriptional profiles could be due to changed proportion of differentiated cells. Similarly, different representation of each population could lead to differences between small and large intestine.

We agree that our approach that is based on cell ablation and precursor cell reconstitution has its pitfalls, which we openly discussed in our manuscript and further expanded on in response to some of the reviewer’s comments. However, we believe it is a valuable complement to emerging sc approaches that infer on developmental pathways by bioinformatic ‘pseudotime analysis’. We believe it is fair to assume that the same populations are depleted and reconstituted in the small and large intestine.

3) Shaw et al., 2018, Schepper et al., 2018, and Schridde et al., 2017, performed transcriptional profiling of these populations during distinct developmental windows or during repopulation after ablation of individual populations. It is unclear how the data presented in Gross-Vered fits with the findings in these publications. In Discussion the authors note minimal overlap with Schridde and claim that differences in finding (which are not elaborated) are because Schridde et al. did not synchronize precursors, but this could be because the ablation strategy removes and does not repopulate a broad range of cell populations.

As we discussed for the example of our comparison to the data by Schridde et al., the use of the distinct platforms and distinct experimental set ups renders direct alignment of data from different studies problematic. We concur with the reviewer, and acknowledged in our revised manuscript, that in our ablation/reconstitution strategy not all cells might be replaced and due to the limited time window not all cells are regenerated. However, we believe that our comparison of small and large intestine reconstitution within the same animals remains valid.

4) A mapping or network analysis would be helpful to understand how these cell populations could be functionally distinct. It is unclear how any described differences would lead to differential outcomes in the tissue. The authors have superficial analysis with listing of GO terms, but these are quite generic.

We thank the reviewer for this suggestion and have added a network analysis using Metascape (Zhou et al., 2019, to the revised manuscript as Figure 1—figure supplement 3B-E.

5) Unclear what the comparison with monocytes adds. These are functionally very different cells so it is not surprising they have distinct transcriptional profile from tissue resident macrophages.

We and others recently noted that Ly6C^-^ monocytes display macrophage features, including their relative longevity and fact that as opposed to classical monocytes, these cells seem not to extravasate. As Ly6C^+^ monocyte-derived steady state cells these cells resemble gut macrophages, although they of course reside in very distinct environments. We performed the comparison of tissue-resident and vasculature-resident monocyte-derived cells to gauge this environmental impact.

6) What do the authors mean that since two of the ileal samples cluster with the colon it means the colon is the default? More data would need to support this statement.

We report in our study 12 samples of engrafted cells retrieved from ileum of recipient. Two of these, isolated at the day 4 time point, displayed a colonic signature. In the Discussion we offer as explanation that differentiation into colonic macrophages might represent a default, but we have now stated more clearly that there might be other options.